# Brain-Related Research as a Support Mechanism to Help Learners to Acquire Full Literacy

**DOI:** 10.3390/brainsci13060865

**Published:** 2023-05-26

**Authors:** Heikki Lyytinen, Natalia Louleli

**Affiliations:** 1Department of Psychology, University of Jyväskylä, Fl-40014 Jyväskylä, Finland; 2BCBL, Basque Center on Cognition, Brain and Language, Mikeletegi Pasealekua 69, 20009 Donostia-San Sebastián, Spain; n.louleli@bcbl.eu

**Keywords:** reading, literacy, digital learning environment, GraphoGame, comprehension game

## Abstract

Possibly some of the most important skills that one can have are those needed to become fully literate. We all wish our children to reach such a goal. Unfortunately, the focus of attention in reading research has been on acquiring readiness to sound out written language, i.e., the basic reading skills. Full literacy is the readiness to learn knowledge by reading. Thus, one has to be able to take two steps to reach full literacy. Indications related to both of these steps can be observe in the brain. This may be easiest when we observe the brain activity of a learner who faces difficulties in taking these steps. In fact, the serious difficulty of taking the first step can be observed soon after birth, shown below as a summary of relevant details from the paper published earlier in this journal. The step from a basic reading skill to reading comprehension requires that one must learn to read for the mediating meanings of the text, i.e., its morphological information, on top of the phonological one. This can also be approached using brain-related observations, as we show here, too. Taking these steps varies between orthographies. Here, we illustrate the learning of these steps in the context of transparently written alphabetic writings by choosing it as our concrete example because its readers form the majority of readers of alphabetic writings. After learning these facts, we had to be able to help those who face difficulties in these steps to overcome her/his bottlenecks. We summarize how we have tried to do that. Each step can be taken using a digital game-like training environment, which, happily, is now open to be distributed for the use of (almost) all in the world. How we have already tried that concerning the first step is illustrated below. Additionally, how we plan to do that concerning the second step, the final goal, completes our present story.

## 1. Introduction

The awareness of the senior author was peaked only after the most important part of the Jyväskylä Longitudinal study (JLD) was completed and when we had begun to investigate and approach the mystery of dyslexia. Our approach started by following children with a familial risk for dyslexia from birth to the age in which the Programme for International Student Assessment (PISA), 15 years old, is taking place. The JLD revealed that it is auditory insensitivity that may compromise the acquisition of the basic reading skill–taking of the first step towards reading. It can be noted using brain measures called mismatch negativity before any environmental effect can have any role, i.e., at the age of 3–5 days of life (see the review of our related studies [1]. After illustrating this in more detail, we move to the description of how we are organizing our operations to also have the second step supported using a comparable logic.

### 1.1. The Brain Basis of Learning the Basic Reading Skill

The aforementioned observations from the brain from a very early age reveal that one has to be able to perceive sound well enough to learn to connect the speech sounds (phonemes) to letters. If the genes (realizing the familial risk for dyslexia) have affected the brain in such a way that even the pitch differences among basic sinusoidal sounds are not handled accurately by the brain, one faces difficulties in acoustically connecting the closest phonemes to the letters representing them. Thus, when we observed how well learners could connect the acoustically closest phoneme sounds of the l, m, and n to their letters, it showed to be very difficult. Happily, it can be made by drilling, which means a large number of repetitions that only can happen in a digital learning environment that we have developed to help all children learn the basic reading skill. The game format of this environment is called GraphoGame. It keeps children engaged long enough to elevate their sensitivity to differentiate the sounds that are relevant for connecting spoken and written units together in such a way that accurate basic reading skill becomes possible to acquire.

The JLD results reveal that spoken and written language skills are tightly connected. Children whose early language skills are developing most optimally are also the ones who first reach the goal of reading, which is not the sounding out of written language but the ability to collect the meaning the written language is mediating to the mind of the reader, and which is most convincingly reflected in PISA-results. Thus, it is necessary to attend the second, last step as intensively as the first step in reading research. This has not been the case until now. The first step has been attended much more frequently, and most of the resources explicitly meant for supporting reading skills in the schools have been spent for the support of taking the first step.

What we made after we learned how to identify children in need of support for acquiring the basic reading skill was the development of a digital training tool that helps children overcome related problems. Today this training tool (see grapholearn.info, accessed on 2 March 2023) is used by millions of children in the world via the GraphoGame company (see www.graphogame.com, accessed on 2 March 2023) after we had documented the efficiency of our technology to be effective in supporting taking of the first step [2].

Below, we will summarize the theory behind this training tool and what is the empirical evidence on the basis of which we believe (and in fact have demonstrated) that it can help effectively also children with the most severe dyslexia. This only happens via very carefully following the guidelines we have given to help the users know how it should be used. This, and the whole theory behind the efficiency of this digital learning, is based on brain research, which opened the mystery of dyslexia as one of the main results of the JLD, as will be shown below.

As foretold, the acquisition of the basic reading skill or taking the first step is not enough. The goal of reading is the mediation of the meaning of the text and learning knowledge by reading. This is, in a sense, the basis of the most important part of our knowledge learning during our schooling. Most knowledge in the school is learned by reading, although guidance on how to do it is given orally by teachers. Unfortunately, it looks like this part of teaching is not always successful, which has been seen in the decline of the results of PISA, which has been observable also in Finland—one of the most literate countries in the world. Interestingly this is true, although newspaper reading is known to be in the top level in Finland. The decline is more clearly observable among boys and men. This is most likely due to a decline in reading activity outside school, as a recent national evaluation of school learning in Finland revealed.

Interestingly, the status of reading and learning described above characterizes people who are reading transparent alphabetic writing systems (read by the majority of users of alphabetic orthographies). This may be because learning to read such writing can be made without any idea of what the text the reader is mediating to her/him. Readers of transparent alphabetic writing can read and sound out whatever comparably written language, such as local African languages, relatively accurately and fluently, while the meaning of the text is not at all comprehensible. We cannot understand absolutely nothing from the text of Sub-Saharan African local languages; however, this fact is not true in most northern African countries. This is not the case with writings in the Chinese language, whose learning starts moving the attention of the reader to the meaning from the beginning. This makes it very easy to understand that at the top level in PISA-based comparisons, we have readers from many countries where the writing systems have such a heavy portion of morphological information.

However, how is the second step taken in fully transparent writing? Most naturally, by starting to read as soon as one has acquired the basic reading skill. Even reading one detective story might make it evident that the enjoyment of such a reading is based on a search for the meaning of the story. This means that one had to learn to emphasize such contents of the written material, which mediates the key information, e.g., in the context of a detective story, as follows: what is told about the potential murderer. As was written in the national evaluation of school learning, reading outside school has a clear correlation to knowledge learning from the schoolbooks, i.e., to school achievements.

The important question that we had to find an answer to is as follows: how to make students who are not interested in reading able to reach the goal of reading? Such disinterest is more common among boys whose preferences are more and more related to computer games, etc., which contain fewer reading-related activities. How to help them and also those who would be interested in reading but do not have any interest in reading material, such as African children? This is our next focus answer, which can also be partially opened by starting from the brain-related observations. 

### 1.2. What One Has to Learn to Take the First Step towards Literacy

In the learning of the basic reading skill, one has to learn the connections between spoken and written language. The learner has to have sufficient knowledge about the spoken language to acquire written forms. None can learn to read a language whose spoken form is totally unknown to her/him. All languages can be learned by applying the association learning principles mentioned above. This theory has been proven empirically in the context of learning to read a number of times in my research. The first documentation showing it was published years ago [3]. Later on, it was expanded upon a number of times, first documenting how applying this principle might help not only Finnish children but also African children [4]. A most critical empirical study documented how it helped children with a risk of dyslexia in Finland [5] and how this could also be shown by observing changes in the brain in Switzerland [6]. At the same time, we were also documenting its efficiency more extensively in Africa [7]. The whole sequence of proceedings showing how it has helped in many countries and orthographies is reviewed in [2]. 

This theory also applies to the learning of English, but no research of anyone else has focused on it explicitly as far as we know. Why English-based research has not ended up connecting reading acquisition to this basic principle of learning has been explained by us recently (Lyytinen et al. submitted). Just to summarize the likely reason as follows: it is difficult to note the connection-building in a writing where none of the vowel sounds are represented by the same letter in all contexts of written English, and even the most consistently behaving letter has not been named in the way that it could cue about this most common sound it represents, which is /i/.

It is interesting that most alphabetic writings have followed the Latin mode because missionaries (who have known it) have wanted to make Bible readable to as many as possible. Thus, languages such as German, Italian, Spanish, and Sub-Saharan African local languages have relatively transparently written orthographies at a grapheme-phoneme level. It is, however, very surprising that missionaries have not even today observed that understanding written language is heavily compromised among the readers. There are fewer problems in instructing people to read the text. However, in Africa, this can happen only by using optimal ways to train them, which has not been available there. Now when they have had effectively trained with GraphoGame, they have shown to learn to read. Because this can happen without one needing to attend to the meaning at all, the full literacy related to local languages is mastered by few only in Sub-Saharan Africa. Much more African readers would master the second step if they could have materials to apply the basic reading skill they have acquired. The schoolbooks are not enough. However, this is not such a big problem because the local books would not cover a sufficient amount of information to help enough. Therefore, the acquisition of some language that has a rich literature that has to be learned. This will start to be learned years later, e.g., in Zambia, which delays children’s opportunities to have access to knowledge, which is why the English GG would be very helpful in making learning to read faster. Happily, it is now available (see www.graphogame.com, accessed on 2 March 2023) for also learning French, Spanish, and Portuguese, which represent common second languages children learn to read in Africa.

In rich countries, the possibility to take the next step by simply reading enough happens relatively naturally among most readers, although during recent years, the PISA results have been declining among boys in many countries. The most likely reason for that is that more and more boys are replacing reading with other activities, such as playing computer games. A most interesting observation is that this was first noted in Finland, a country where the likelihood that one is interested and willing to read newspapers is at the top level in global comparisons. When we started writing this article, the national body nominated to assess school learning in Finland had just published its data, revealing a substantial decline in school learning. This happened after three subsequent operations of PISA concerning literacy had shown a continuous decline, especially among boys. 

### 1.3. How to Open the Difficulties in Taking the First Step–Basic Reading Skill 

We mentioned that the empirical support of the theory we propose to explain the acquisition of basic reading comes from the way we have successfully instructed the basic reading skill via training with the GraphoGame learning environment, the learners to identify the connections of spoken and written language. This has happened by motivating the learners in a game-like digital environment (GraphoGame) to identify the letter (among a number of alternative letters) each of the phonemes represents when they hear from headphones each of them. Each spoken item (including larger units, syllables, and words) is given, in turn, to connect to the corresponding written item (among alternative ones). This very same method works in all written languages, but in English, the starting items have to be larger units to instruct only connections that are learnable, i.e., always true.

The JLD study revealed that among children who have a familiar background of reading disorder, the likelihood of facing difficulties in learning to read is about fifty-fifty–almost half of them face problems. If the spoken language is delayed, it increases the risk. However, happily, the real risk can be easily observed by asking the child to name letters before school age. If the child is unable to name most of the letters correctly at the time of their school entry, they must immediately start receiving preventive practice. 

Among at-risk readers, successful learning only happens if the GraphoGame is used preventively. After learners have faced severe problems in learning the basic reading skill, their experiences have been so bad that it has elevated their avoidance behavior up to the level where any training is less effective help, and many are no more willing to approach even in this way the experience they had when they tried to learn but failed. Few, if any, teacher(s) are/is able to provide long enough positive face and feedback to keep the bad experiences under control. The preventive use of the GraphoGame helps to avoid such experiences because it is a very tolerant trainer that does not provide negative feedback in such a way that it could be experienced as something intolerable. By collecting the history of learning, it can offer such connections to be responded to, which the learner for sure knows after a few incorrect responses, thus avoiding producing unpleasant experiences for the learner.

One of the surprising things is that when the connection building between spoken and written needs larger units in learning to read English to instruct connections that are “learnable”, (i.e., true at least mostly), we find that often the most well-working units can be whole words. In environments with transparent alphabetic writings, most people prefer learning to read also English. Because the phonics way to teach would require mastery of the phonemes of English that most teachers do not know, the practical solution to instruct spoken English has jumped to the use of whole words. Thus, children learn English like one learns foreign words by looking at the appropriate dictionary, which is a nice way to learn the vocabulary of English. This naturally can work for learning to speak English if someone can help how such words can be correctly sounded. For the purposes of learning to spell English, such an approach works so well that learners not only store the orthographic images of the English words but also become able to spell English more accurately than many native English speakers do. Thus, the accidental problem of teachers instructing the phonemes of English, which very few master, has ended in most non-English speaking countries using this whole word approach for not only learning vocabulary but also acquiring the basic reading skill of English.

To be able to sound English words accurately, one has to have some idea of the meaning of the words, i.e., have some morphological information (i.e., smallest meaningful units of the language), and this may help a little bit in taking the second step–to understand what the words mean. This is quite different from what one learns when acquiring the basic reading skill of transparent writing, where one needs to learn only the sounds without having any access to the morphological level of the language to be able to make the first step. This is why it is more demanding for non-English speakers who have learned to read a transparent writing to learn to reach the goal of reading, i.e., mediate the meaning of the words and sentences while learning to read. Naturally, a higher-level comprehension is needed for learning to read knowledge, which is a little bit more demanding from the starting point of mastering transparent writing. However, in practice, readers of both transparent and nontransparent writing have to read a lot or find another way to learn to comprehend the written language as well they can understand spoken language. However, for the learners of transparent writing systems, the second step is most likely more demanding.

### 1.4. What Happens in the Brain when One Tries to Make the Second Step?

Language acquisition requires complex cognitive skills. Infants’ initial steps in language acquisition involve acquiring phonemic and phonological representations. From the moment an infant is born, he/she is capable of discriminating phonetic contrasts, which by age seem to be driven by the linguistic patterns of his/her native language [8]. As time passes by, infants are able to discriminate more efficiently the phonetic contrasts belonging to their native language compared to the ones that do not belong [9]. When reaching twelve months, each baby has already acquired many vocabulary units stored in the lexicon [10]. Morphological awareness also appears to start very early in life. A study investigating morphological awareness in 15-month-old babies demonstrated that babies were able to decompose a word into word stem and suffix in the Hungarian language, a pattern that occurred only when the presented words contained a highly frequent suffix [11]. The aforementioned research was performed with a behavioral measure (head-turn preference paradigm). This is one of the earliest indications (15 months old) of the gradual development of morphological awareness in humans [11]. Morphological awareness develops continuously from infancy to young children’s age by building up the morphological representations of their language, which are mainly represented as rule-based mechanisms, i.e., young children have the ability to correctly inflect words [12] or process morphosyntax in sentences [13,14,15]. 

Morphological awareness is the ability to associate the roots/stems of words with the meaning of each morpheme, and a speaker should be able to identify and manipulate morphemes [16]. Inflection, derivation, and compounding are the three morphological processes taking place in most languages; their function is to accordingly change the morphemes (the smallest units of language with meaning) in order to change grammatically a word (inflection) and/or to create new words (derivation) and/or to mix two existing words (compounding) [16]. A study by Casalis and Luis-Alexandre in 2000 showed that morphological awareness (the ability to recognize and manipulate morphemes) gets better with age [17]. In their research, they tested longitudinally French-speaking children while performing derivational morphology tasks from kindergarten to second grade [17]. Interestingly, in another study by Lyytinen and Lyytinen in 2004, inflectional morphology was tested with tasks measuring children’s performance with inflectional morphology, and they found that children develop morphological awareness for inflectional words between the ages of two and four years [12]. 

Furthermore, morphological awareness was found to play an important role in the decoding of words in English, Finnish, and Chinese. Specifically, in the study of Nagy et al. in 2006, morphological awareness was highly correlated with vocabulary reading for school-age children, especially for 4th and 5th grade and word reading and reading comprehension [18]. In line with these findings, in a study by Kirby, Deacon et al. in 2012, morphological awareness was significantly correlated with word reading and reading comprehension [19]. Similarly, morphological awareness tested in Finnish children during 1st grade of schooling was found to be highly correlated with reading comprehension [20]. Finally, similar associations between scores in morphological awareness tasks and word reading and comprehension were also found in Chinese, suggesting that the association between reading development and morphological awareness is not language specific and observed across different writing systems [21]. Thus, morphological awareness seems to be very important for reading acquisition, and it is found to predict later reading skills in children during the first, second, and third grades [19]. 

Research investigating morphological acquisition and awareness has mainly focused on morphological operations across different languages depending on the type of linguistic system. Derivational morphology results in the production of more words compared to inflectional morphology because derived words can allow for larger changes in meaning compared to inflected words [22]. Interestingly, the processes of derivational morphology can be recurrent, as each speaker can add more than one morpheme per word, i.e., “un-happi-ness” for the production and/or creation per stem/word. During the processing of deriving stems/words the morphemes get combined with a stem/word, the ones which are attached at the beginning of a stem/word are called prefixes (i.e., un-happy), whereas the ones that are attached after the stem/word are called suffixes (i.e., play-er). In the Greek language, there are morphemes that are attached in the middle of a word, which are called infixes, but they are not used in every language. As a general rule, the derivational morphemes change the definition of a stem/word, i.e., the prefix -un changes the meaning of many adjectives in the English language by giving them a negative connotation. 

Event-related potentials (ERPs) and event-related fields (ERFs) were the brain measures used to study the awareness of derivational morphology in many populations (adults, adolescents, children [23,24,25,26,27] as well as with event-related fields (ERFs) in adults [28,29] and in children [30,31,32].

Morphological awareness consists of awareness of inflectional and derivational morphology, which both play an important role in reading acquisition [16]. Interestingly, the processes of inflectional morphology can change the grammatical information in a word/stem, such as number, i.e., singular or plural, by adding a suffix (e.g., English plural number [boy-s] or tense English past tense [open-ed]. 

Event-related potentials (ERPs), event-related fields (ERFs), and blood-oxygen-level-dependent contrast (BOLD in fMRI) were the brain measures used to also study the awareness of inflectional morphology [27,33,34,35], event-related fields (ERFs) [27,36], and fMRI studies [37,38,39]. 

Learning to read accurately in the Finnish language or, e.g., local Sub-Saharan languages does not necessarily require any previous morphological awareness of the language since their phonetic/phonological system is very transparent (one-to-one grapheme to phoneme) [40,41]). Despite their very transparent phonetic/phonological system, some languages have a complex morphological system with rich inflectional and derivational morphology (Finnish: [42]). Previous studies measuring the behavioral performance of young children have shown that awareness of derivational morphology is correlated with accurate word reading, especially in languages with transparent orthographies and rich morphological systems (i.e., Italian: [43]; Spanish: [44]; Greek: [45,46]). 

As far as the comprehension game is concerned, this learning environment is focused on training the learning and comprehension skills of children who already know how to read. The fact that comprehension game includes sentences, and these sentences include words that bring morphological processing even closer to comprehension. As morphological processing is one of the linguistic processes (phonology, syntax, semantics, pragmatics), it is closely linked to comprehension skills used/trained via the comprehension game.

### 1.5. What Is Needed to Make the Second Step towards Literacy

Learning to comprehend written language is a highly complex cognitive process; however, it has some relatively clear and self-evident features that are open to being instructed without complexities. The starting point is that one can comprehend such content of written language for which the learner has appropriate background knowledge. Thus, if we look at how school learning happens, children are guided to learn content step-by-step. This means that the background knowledge of whatever new issue has to exist before one is able to fully comprehend a new issue. Thus, school learning is a good context for training reading comprehension. 

Moreover, the requirements associated with comprehension should be the same concerning spoken and written language. This requirement is again fulfilled if we attend school learning because it is the teachers’ duty is to guarantee that the words used in the written lessons should be known by the learner. It is also interesting that natural comprehension from written language can be easier than from spoken language due to the possibility to use more time with the text than with spoken knowledge, which is not always given in a sufficiently repeated way. Text allows whatever number of repeated readings, which can help in attempts to understand what the text means to tell its reader. 

The strategy supporting reading comprehension, which is relatively easy to apply, is that the learner who wants to follow the meaning of written language has to be trained to have the readiness to categorize words according to their importance for following the red line of the story. Not all words are equal. Some are more important for understanding even a sentence and more so if one tries to understand larger chunks of the written language. This choosing process is relatively easy to train, but it takes time. This is why it is good to start training from the time one has taken the first step to be able to read accurately and fluently enough, i.e., the basic reading skill. This means that in most environments of fully transparent writing, this training can be started during the second grade when learners master the basic reading skill fluently enough.

The key bottleneck of learning to comprehend written language is that the working memory has a limited capacity. Thus, it had to suffice to keep such an amount of information active, which makes it possible to follow the red line and, at best, to integrate large amounts of information if the goal is to understand complex knowledge-intensive content. Typically, this means that the learner had to be able to keep in the working memory only a few points-mostly only the key points of a few sentences per page of a book to be able to follow the red line. 

The most efficient readers—so-called speed readers—are efficient because they have trained themselves to find fast the important bits of information from the page to build the story on the basis of that which can be maintained in the working memory for long enough. This can make them much better learners than typical readers.

## 2. Conclusions: How We Plan to Open the Readers’ Bottlenecks Associated with the Development of Comprehension Skills after They Master Reading

As already briefly mentioned, earlier the practicing of reading comprehension may happen most naturally by making it in the context of learning lessons from schoolbooks. This means that teachers had to be trained and motivated to implement the key content from the books to the comprehension game (CG), which is then used by pupils to complete their reading of the lessons. If children are first reading the lesson from the schoolbook and then confirming that they have really noted all the important contents, they can do it by moving to use CG after they read the lesson. This has several benefits. Naturally, the main benefit is that pupils slowly learn how to find the key contents from whatever material. A more immediate result is that the lesson will be learned more deeply and faster. A further benefit is that teachers can follow the proceedings of how pupils learn their lessons to guide their instruction in a spoken way to issues that seem not to be easy for the learners to acquire from the lesson. The possibility of following learning can, in the end, become elaborated to replace exams. In this case, it happens in more beneficially because the CG applies dynamic assessment (DA). DA means assessment of the readiness to learn identifying where it still faces bottlenecks, i.e., where the learner has not yet reached her/his goal. It reveals where the instruction has not yet helped the pupil well enough and shows thus also to teachers where their instruction needs to be more effective and sensitizes her/him also to individual needs covering separately each of the children of the classroom.

In fact, CG opens a way to individualize instruction to adapt to the needs of support each child has to receive to be able to follow the curriculum. It needs then not only that they use more time but also that they get what they had to learn in an easier-to-acquire way. Mostly the main problem is not learning itself, but the engagement–readiness to expose her/himself to training. CG may offer a more engaging way to also learn difficult content with its game-like instruction. We are doing our best to make the CG as gamified as possible to make it also tolerable to boys who may not otherwise be as willing to train themselves to read with comprehension, which would need a lot of reading outside school.

One of the challenges is the one associated with motivating teachers to take the role they have in this type of instruction. Because teachers are in a key role, they have to let themselves be trained for good “teacherhood”. A good teacher knows how and what children can learn and what the pupils at each stage need to be instructed to learn. This means that teachers had to become able to identify the key content from whatever schoolbook they are obligated to teach. Succeeding in making the proposed content interesting enough to keep children trying for a sufficiently long time for storing the key content. This trains the teachers to understand how much it may help them in their work. 

In the end, most of the knowledge-related learning of the pupils can be moved to CG to do. When the content of a schoolbook is implemented, it will be open to being used for so long time then this book is used. This naturally depends on how many years the teacher is instructing such grade for which s/he has made content. It is also possible to accept implementations of the key content that other experts have performed. For example, in Finland, the developers of the CG have implemented the contents of all the books available for the second graders from different publishers to the game. 

To understand how the implementation of the content to CG is made, the following short description may be sufficient. The principle is that somehow the key contents had to be read actively so many times that all is for sure become stored in the permanent memory of the learners. Making it in a way it is performed in the CG adds one interesting benefit at the top of those listed above. This is critical reading. If some portion, e.g., 20–30% of the key contents is not true, which the learners have to identify to reach the goal of the game, it keeps them repeating long enough because the game is over only after all claims/sentences have been responded correctly as being true or not. 

The key contents themselves are given as claims such as “Helsinki is the capital city of Finland”, picked from geography lessons of the second graders’ schoolbook about countries of the world and, in this case, concerning Finland. Thus, providing a set of such sentences (claims), which summarize the key content of a lesson, which also include claims that at best are misunderstandings (not-true) the learner may have, and ask her/him to choose in turn each sentence whether it is true or not. Most learners have to repeat making judgments several times all the sentences before these are all chosen without any errors to get the game to reach its end. This way one can elaborate the claims to a form that provides repetition sufficiently many times for reliable storing of the content players had to learn. It is important that the storing covers not only the true contents but also get rid of misunderstandings. 

If pupils start reading their lessons first in the traditional way and then using the comprehension game from the grade learners, they have a fluent basic reading skill to be able to play the game, they most likely become quite good in comprehension of written language early enough to start learning their schoolbooks efficiently at time. Finally, after several years become able to succeed well in PISA. This all is what those children who are not interested in starting to read outside school are needing. 

As mentioned, it is most often boys who need this new way to learn knowledge by reading because they are no more so interested in reading outside school as they used to be. This trend has already lasted for many years because the adult men, even in Finland, end up with poor skills in the comprehension of written language because they have not read enough. It is surprising that the portion of such poor comprehenders is larger in Finland than the portion of children for whom most resources are spent for helping them to take the first step. This is even more surprising in those countries (such as Finland, France, the UK, and the USA) where the training tool for the basic reading skill, GraphoGame, is made available. As described above, with appropriate use of this tool, everyone had to learn the basic reading skill to then start using the comprehension game for reaching full literacy.

### Final Summary

The final conclusion obligates us to give the following definite answer to the most important question “How have our brain-related observations helped us to define means to train children to overcome the reading-related problems until the end?”: supporting children to acquire readiness to acquire knowledge by reading, i.e., reaching the goal of reading?

It is relatively safe to conclude that at the top of the very good success of our training environments for guiding children to full mastery of the basic reading skill independent of the problems they have with reaching it, our brain observations—although are still only preliminary due to small sample-size—motivates us to conclude that our approach to defining reading acquisition as a connection building operation between spoken and written language is a valid approach. This is because the conclusion of defining the most serious problem of reading—dyslexia—to be a result of auditory insensitivity is also seen during our training results in the context of the use of GraphoGame. Those connections that are most difficult to store reliably are those between acoustically closest items, such as those represented by the letters of l, ma, and n. After these have been successfully learned, the most difficult bottleneck children with dyslexia are facing has been opened, and this succeeds only after long drilling.

The second challenge we defined to be facing is the reaching of full literacy is the way children can reach it. The natural way is reading a lot. Unfortunately, this does not happen today anymore as likely it used to earlier. Especially boys have become more and more inclined to read enough to start reading the goal of reading. The training they need must be as direct as possible but such that it engages them more likely than interesting reading material can do. Our solution is to motivate them to use the following same approach they prefer when choosing who is the winner: use time for playing computer games or reading. The natural way is to use the following approach: computer games. Additionally, helping it to win in getting their attention by providing an offer that is difficult to deny because it saves their time for the use of games that they prefer the most. This can happen if our game helps them to save time needed for learning the lessons and not only save time but also receive positive feedback from the teachers by supporting them to achieve better in the school concerning the reading-based contents of school learning.

We may have to confess that our brain-related observations may be helping less in making our training solution to train learners to reach the full literacy more convincing. However, at least we have made our best to try to find brain-related support for our work for supporting literacy acquisition getting more visibility among the many people to whom brain-related evidence is more convincing than that resulting from behavioral observations.

## Data Availability

No new data were created for this study.

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
