# Peer review of "Brain-Related Research as a Support Mechanism to Help Learners to Acquire Full Literacy"

_brainsci, 2023, doi:10.3390/brainsci13060865_

Round 1
Reviewer 1 Report
The manuscript presented here is really interesting and well written.
Authors provide a comprehensive review of the literature.
However I would suggest some minor modifications, also please read carefully the manuscript there is some typo.
I would suggest to provide more background in the introduction and clearly state why you write this review.
In most of the sections it misses citations, please add citations.
Section 5: There is "5…by Natalia" you should delete it
Section 6: "training the from the time one has just taken the first step" typo again
Author Response
Thank you very much for all your thoughtful comments!
We appreciate your effort in checking our manuscript!
The nature of our writing has been improved a lot and mistakes have been edited/modified accordingly. All the typos have been fixed/removed accordingly.
As far as the citations are concerned, this selective review is based on tens of publications (mainly the senior author’s own publications), most of which are typical empirical studies, but some reviews of the tens of papers based on how we have learned to know practically better than most other literacy researchers about reading acquisition. Thus, we have learned extensively how to help children and teachers in using our digital tools from which the ComprehensionGame is something which none else has even tried to develop.
Reviewer 2 Report
This is a nice manuscript. At the same time, it's more of an "Opinion Letter" than a "Review", as the ms talks mainly of the tools developed by the first author rather than a comprehensive review of the literature. I was not convinced bu the title either "How brain-related observations can be used to support learners to acquire full literacy?". It should be changed in a revised version of the manuscriot.
My advice is that if the authors want to stress the virtues of GraphoGame and ComprehensionGame, it is fine, as long as they are a "Conflict of Interest" section, as this is proprietary software developed by the first author. Also, the Opinion Letter should be written in a much shorter manner than it is now, mainly articulating past and recent research in Finnish and other languages at various school levels.
As these changes require a profound manuscript rewriting, making it shorter, I won't make more suggestions at this stage. I must say I was a bit confused with the initial paragraph of the ms, and I advise the authors to modify it a bit.
Some figures of how the software packages work or some evidence across transparent versus deep orthographies would also be welcome.
There are a few other more formal questions, such as the inter-line spacing, that changes every few paragraphs, and also some proofreading is necessary as on section 5 "5. What happens in the brain when one tries to make the second step? 5…by Natalia" I guess "by Natalia" should not be there.
Author Response
Thank you very much for all your thoughtful comments!
We appreciate your effort in checking our manuscript!
The nature of our writing has been improved a lot and mistakes have been edited/modified accordingly. All the typos have been fixed/removed accordingly.
As far as the title and the type of our manuscript, we changed it accordingly, so that it fits better with our content. Our manuscript is now considered to be published as an Opinion letter. We also edited the length of our manuscript, it is now shorter in length, but still very comprehensive.
We also fixed the style of the manuscript (inter-paragraph spacing and numbering) as well as section 5, which has been modified/edited as suggested in order to be written in a more comprehensive way.
Round 2
Reviewer 2 Report
There are still a few issues that could be improved:
(1) the first sentence in the Introduction is unnecessarily long. Furthermore, I we haven't solved the mysteries of dyslexia (perhaps "beginning to approach" or something like that, not so drastic as it is now).
(2) the type of paper should be Opinion in the text (it still appears as Review)
(3) Section 2 should be divided into subsections, and then the authors need to add a final paragraph summarizing their main message.
Author Response
Thank you for your comments!
The first sentence whas been modified, so as to be more comprehensive, less drastic and shorter, as it was recommended.
We changed the remaining parts of the text, where it was appearing to be a review rather than an opinion, but the modification of the paper's format will be further modified by the journal.
We added a final part, where we summarize the main message of our manuscript.